


**Evaluation of global seismicity along Northern and Southern hemispheres**
**Olaide Sakiru Hammed[1], Theophilus Aanuoluwa Adagunodo[2,*], Musa Oluwafemi Awoyemi[3],**
**Joel Olayide Amosun[4], Tokunbo Sanmi Fagbemigun[4] and Tobi Ebenezer Komolafe[1]**
[1]Department of Physics, Federal University, Oye-Ekiti, Nigeria
[2]Department of Physics, Covenant University, Ota, Nigeria
[3]Department of Physics, Obafemi Awolowo University, Ile-Ife, Nigeria
[4]Department of Geophysics, Federal University, Oye-Ekiti, Nigeria
* Corresponding author e-mail: taadagunodo@gmail;
theophilus.adagunodo@covenantuniversity.edu.ng
**Abstract**
An earthquake has been identified as one of the major natural disasters that cause loss of lives and
property. To mitigate this disaster, knowledge of global seismicity is essential. This research is aimed
at evaluating the Gutenberg Richter b-value parameter and focal depth distribution of earthquake
parameters to identify the prominent earthquake-prone zones in the Northern and Southern
hemispheres. The study area covers 20° to the Northern and Southern hemispheres, with the equator
in the middle. The data were obtained from the earthquake catalogue of the Advanced National
Seismic System (ANSS) hosted by the Northern California Earthquake Data Centre USA from 1963
- 2018. Fifty-four-year earthquake data of $M \geq 6.0$ were processed and analyzed using Gutenberg-
Richter (GR). The b-value parameters obtained from the GR model were plotted against the
hemispheres using bar chart graphs to determine the tectonic stress level of the study region. The
earthquake energy released was evaluated along the Northern and Southern hemispheres for a proper
understanding of seismic events in the study region. It was observed that the rate of earthquake
occurrence at the Southern hemisphere is higher than the Northern hemisphere. The b-values
obtained in all the zones vary from $0.82 - 1.16$. At the same time, the maximum earthquake energy
of $4.6 \times 1025$ J was estimated. Low b-values indicate high tectonic stress within the plates. The large
tectonic stress accumulation around the equator suggests that unstable lithospheres characterize this
zone.
**Keywords:** b-value, Seismicity, Focal depth, Lithosphere, Equator, Earthquake energy
**1. Introduction**
The destruction associated with large earthquakes is one of the primary motivations for studying
seismicity. The major concern about an earthquake is its radiative energy being released during its
occurrence, which poses a threat to humankind. Proper investigation of the energy released by great
earthquakes could proffer insights on the dynamics of the lithospheric plates. It has been established
from previous studies on latitudinal variations of the energy released by earthquake events and its
occurrence pattern that significant seismic events were observed to occur below latitudes $\pm 45°$
(Varga et al. 2012; Riguzzi et al. 2010; Levin and Sasorova, 2009; Dennis et al. 2002; Levin and
Chirkov, 2001; Shanker et al. 2001; Varga, 1995; Sun, 1992; Chouhan and Das, 1971). The primary



mechanism that is associated with the release of energy at a deeper source is an earthquake (Kirby et
al. 1991; Abe and Kanamori, 1979; Richter, 1979; Chouhan and Das, 1971). It has been documented
that when these deep energies are released, they are not uniformly distributed along the earth's radius
(Gutenberg and Richter 1956, 1954, 1942, 1936). About 25% of the global earthquakes occur at focal
depth > 60 km (Frohlich, 2006), and of the total energy released, 0.2% came from foci earthquakes
with depths > 300 km (deep focus) (Abe and Kanamori, 1979). The analysis of great earthquakes
from seismically active zones showed that the earth is silent at depths beyond 680 km, and the
deformations at these depths exist within the viscous regime (Abe and Kanamori, 1979). This study
aims to determine the levels of tectonic stress accumulation along the Northern and Southern
hemispheres to identify the regions that are likely prone to major earthquakes in the future.

**2. Materials and methods**
For this study, the data were extracted from the earthquake catalogue of the Advanced National
Seismic System (ANSS) hosted by the Northern California Earthquake Data Centre U.S.A in a
readable format. The data comprised earthquakes of M $\geq$ 6 that occurred along the latitudinal
boundary of 20° S to 20° N (Figure 1) from 1963 to 2018. The data comprised the date and time of
occurrence, the latitude and longitude of the epicenter, the depth, the magnitude designation, source
codes, and event identification. The data were sorted and filtered in preparation for further processing
using the CompiCat software (Kossobokov et al. 2011). The study area, which is a strip of width $40^0$
around the globe with the equator at the middle, was subdivided into four regions, each of $5^0$ widths
along the Northern and Southern hemispheres. The areas along the Northern hemisphere are:
latitudes $0^o$ to $5^o$N, $5^o$ to $10^o$N, $10^o$ to $15^o$N, and $15^o$ to $20^o$N, while the regions along the Southern
hemisphere are: latitudes $0^o$ to $5^o$S, $5^o$ to $10^o$S, $10^o$ to $15^o$S, and $15^o$ to $20^o$S, respectively.

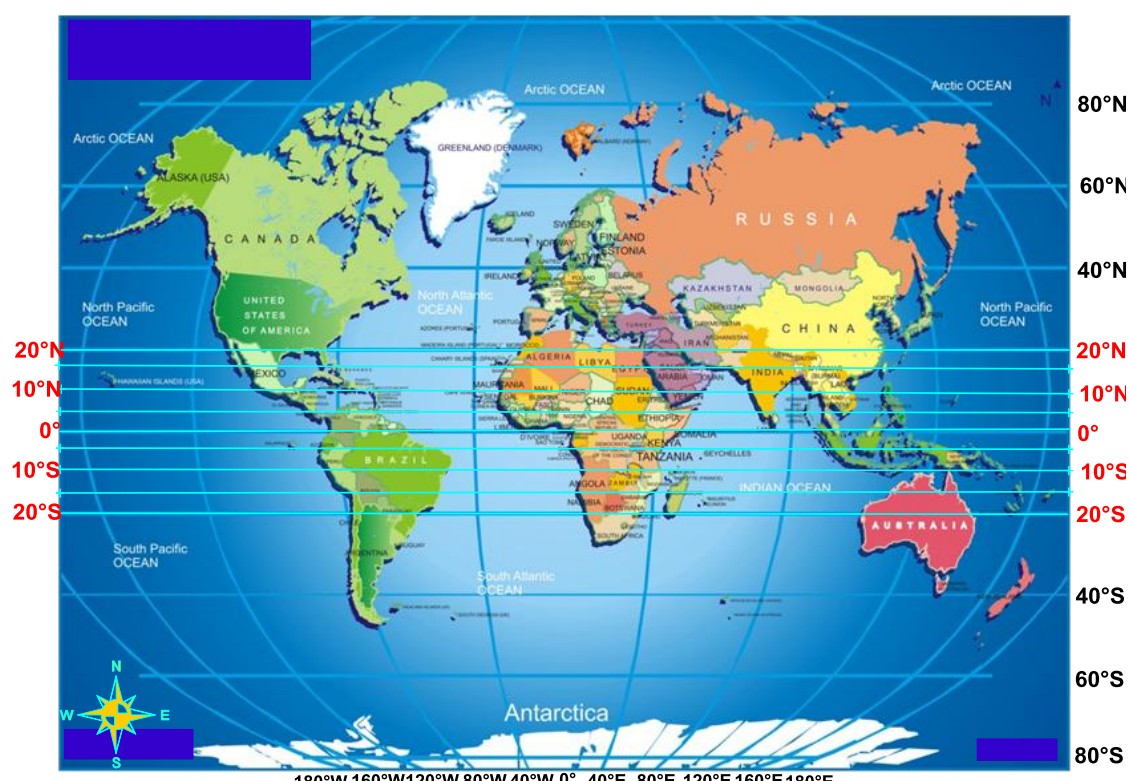


Fig. 1: World map showing the latitudinal boundary of 20° S to 20° N (Adapted from MapsNworld,
65 2011).


The Gutenberg Richter (GR) relation parameter b-value was evaluated in order to establish the levels
of tectonic stress accumulation along the northern and southern hemispheres (Adagunodo et al.
2018a; Hammed et al. 2016a; Nuannin, 2006; Damanik et al. 2010). GR relation is the frequency
magnitude distribution that defines the total number of earthquake events in an area with respect to
their magnitude (M). The GR model used is revealed in Equation (1).
$$\log N = a - bM \qquad (1)$$
where $N$ is the cumulative number of earthquakes with magnitudes equal to or greater than $M$, and $a$
and $b$ are real constants whose values vary in space and time.

An a-value represents the seismicity level, while the b-value corresponds to the slope of the power-
law and describes the size distribution of events. A high b-value indicates a larger proportion of




small events and vice-versa. Also, the b-value represents the tectonic character of an area, which is a
function of the accumulated stress within the lithospheric plates (Hammed et al. 2016a). These two
parameters have been used in the tectonic analysis, earthquake prediction, seismic risk analysis, and
seismicity study across the globe (Adagunodo et al. 2018b). As described by Hammed et al. (2016a),
b-value, which is inversely dependent on the differential stress, and serves as an indicator of stress
level, that is, a region with high-stress level often have relatively low b-value. The b-value of each
section (that is, 5° divisions) within the study area is computed as a slope of the line of best fit.
Besides, b-values are plotted against the hemispheres using bar chart graphs to understand the
tectonic stress level of the study region.
The seismicity parameters such as earthquake energy, earthquake magnitude, earthquake focal depth,
and earthquake epicenter were analyzed. As reported by Amiri et al. (2008), it is paramount to
investigate the seismicity parameters in order to mitigate its effect in the future. For a proper
understanding of the seismic activities in the study area, the energy released by an earthquake is
plotted against its epicenter in the Northern and Southern hemispheres (Ghosh, 2007). The energy
released, E, by an earthquake, can be estimated based on Gutenberg and Richter relations (Gutenberg
and Richter, 1954; 1956), as shown in Equations 2 and 3.

94            $\mathrm{Log\ E = 11.8 + 1.5\ Ms\ (erg)}$                                    (2)

95            $\mathrm{Log\ E = 5.8 + 2.4\ Mb\ (erg)}$                                     (3)

where Mb and Ms are the body-wave and surface-wave magnitudes,
$1\ \mathrm{erg} = 10^{-7}$ Joule (Hammed et al. 2016b).
The variation of earthquake focal depth with magnitudes in the hemispheres was plotted using 3D
CompiCat plots to assess the seismic hazard along the hemisphere on a global scale. The temporal
distribution of seismicity in the hemispheres was evaluated using frequency-time graph analysis.

## 3. Results and discussion

*3.1 Trend of Seismicity and Tectonic characterization along the hemispheres*

104       The degree of correlation coefficient ($R^2$) of the Gutenberg-Richter equations obtained for all
latitudinal zones (Fig. 2 – 9) along the hemispheres is significant enough, due to its closeness to 1, to
serve as a correlator. This implies that both the number of events and the corresponding magnitudes
are much correlated. Thus, the Gutenberg -Richter b values of earthquakes obtained from this
correlation are valid to be used as the tectonic stress accumulation indicator along the hemispheres.
The b-value is the stress meter that indicates the level of tectonic stress accumulation in the
earthquake-prone zones, Hammed et al. (2019). The lower the b-value, the higher the tectonic stress
and vice-versa (Awoyemi et al. 2017; Hammed et al. 2013). In this work, the b values obtained in all



the zones vary from 0.82 – 1.16, as shown in Fig. 2 - 9. Along the Northern and Southern
Hemispheres, the latitude zones 0 - 5N and 0 - 5S that is $10^o$ across the equator are characterized by
very low b values (Fig. 10). The implication of the low b values associated with these zones is that
the zones are embedded with large tectonic stress accumulation. Thus, there is a strong plausibility
that the zones may be prone to disastrous earthquakes in the future.
The analysis of the trend of seismicity along the hemispheres revealed that the rate of earthquake
occurrence in the Southern Hemisphere is higher than that of the Northern Hemisphere (Fig. 11).
This implies that the phenomena that are associated with seismicity, such as the rate of divergence
and convergence of lithospheric plates, are higher in the southern part of the equator. The seismicity
is heavily distributed around the equatorial region at $-5^o$ to $+5^o$ (Fig. 12). This further confirms that
the high rate of tectonic stress accumulation around the equator is indicated by low b-values.
The weighted sum of earthquake energy counts along the hemispheres is densely distributed in the
equatorial region (Fig. 12). At latitude $-5^o$ to $+5^o$, it is observed that the earthquake energy increased
significantly up to the peak of 5.4 x $10^{24}$ J. This is an indication that the equatorial regions are
characterized by large tectonic stress accumulation which implies that the lithospheric layers in these
regions are very unstable.

*3.2 Distributions of focal depth, frequency, and energy of earthquake released*

The focal depth distributions (70 km depth classes) of the number of events (frequency) and
energy of earthquakes are shown in Fig. 13 and 14, respectively. The number of events decreases
with an increase in focal depth up to about 300 km. From 300 to 500 km, the numbers of events are
sparse and sporadic. This zone could be interpreted as a subduction zone (Riguzzi 2010). Beyond this
focal depth range, there is a significant increase in the number of events up to 700 km. Most of the
elastic energy radiated by deep events are concentrated in the depth interval between 500 km and 700
km. In summary, 94.16% of the events occur within 300 km, while the remaining 5.84% occur from
300 to 700 km. About 77.38% of the total number of earthquakes is concentrated in the first depth
class (0–70 km); 16.71% occur from 70 to 300 km; the remaining 5.92% up to 700 km. As regards to
the earthquake energy, 85% is dissipated within 300 km, the remaining 15% from 300 to 700 km,
where most of the energy dissipation is viscous. About 70% of the total amount of energy is
concentrated in the first depth range (0–70 km), with maximum sharp energy of approximately $4.6 \times$
$10^{25}$ J; 15% is dissipated from 70 to 300 km, and the remaining 15% up to 700 km. Only 5.92% of all
the earthquakes released accounts for about 15% of the total energy release at depth 500–700 km.
The deep event clustering is marked by high energy release (energy peak $\sim 0.2 \times 10^{25}$ J).
The predominant clustering of magnitudes M6+ within the shallow focal depth ($< 100$ km) along the
hemispheres (Fig. 15) indicates a high level of stress in the lithosphere around the equator. The



maximum energy of about $4.6 \times 10^{25}$ J generated at shallow depth within this zone suggests that this
zone is characterized by unstable lithosphere.

*3.3 Temporal distribution of seismicity along the hemispheres*

The analysis of the temporal distribution of seismicity along the hemispheres (Fig. 16) revealed
that the earthquakes increase with time, and decade wise. The earthquakes are densely distributed
around the equator in the last three decades. The increase and densely distribution of earthquakes
along the equatorial zones is an indication that the lithospheric layer embedded in this zone is not as
stable as early thought. This deviation could have been due to improved monitoring of the equatorial
region. The increase in tectonic stress accumulation around the hemispheres could trigger continuous
occurrences of an earthquake.




Fig. 2 Frequency Magnitude Distribution of Earthquakes along Northern Hemisphere $0^{o} - 5^{o}N$


LOG N = -1.122M + 9.165
R² = 0.970

Fig. 3 Frequency Magnitude Distribution of Earthquakes along Northern hemisphere $5^{o} - 10^{o}N$



LOG N = -1.167M + 9.476
R² = 0.983



Fig. 4 Frequency Magnitude Distribution of Earthquakes along Northern Hemisphere $10^{o} – 15^{o}N$





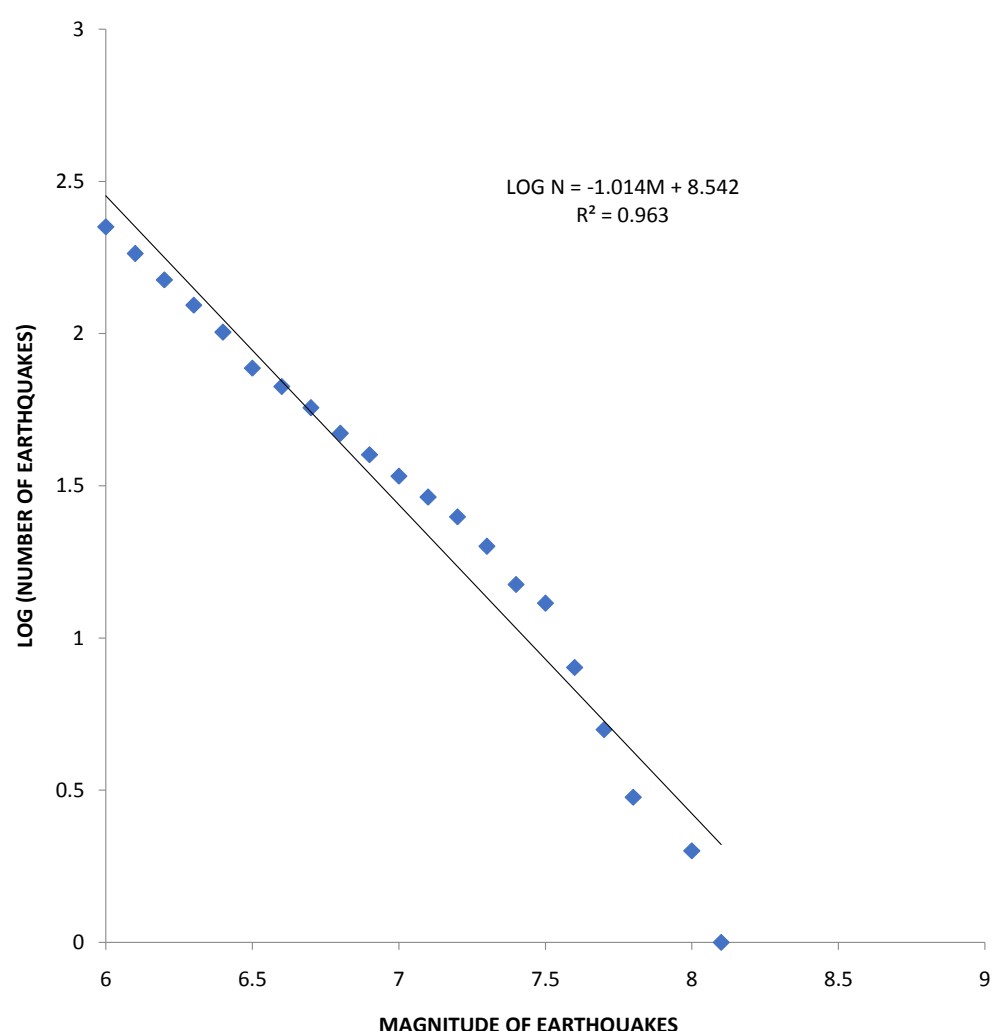


Fig. 5 Frequency Magnitude Distribution of Earthquakes along Northern Hemisphere $15^o - 20^oN$






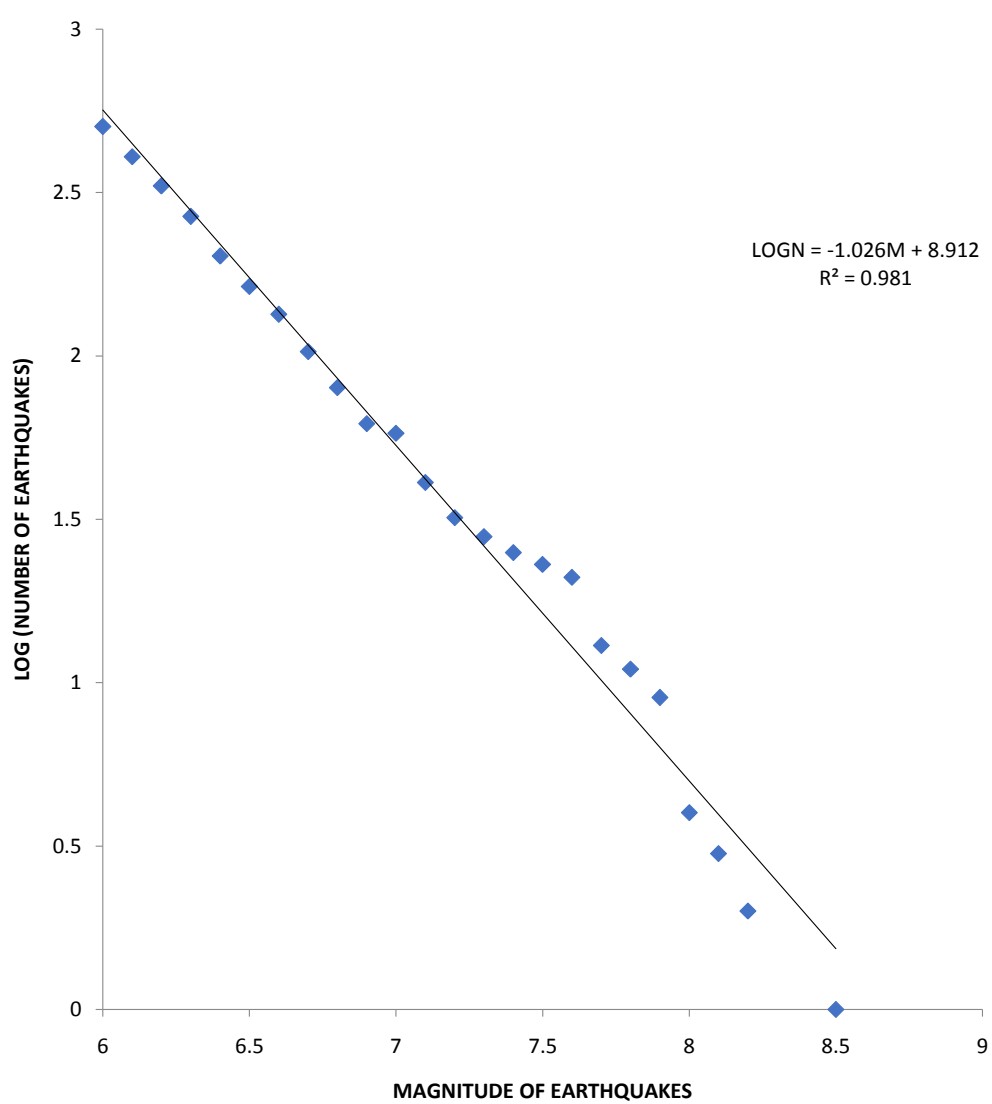


Fig. 6 Frequency Magnitude Distribution of Earthquakes along Southern Hemisphere $0^o - 5^oS$


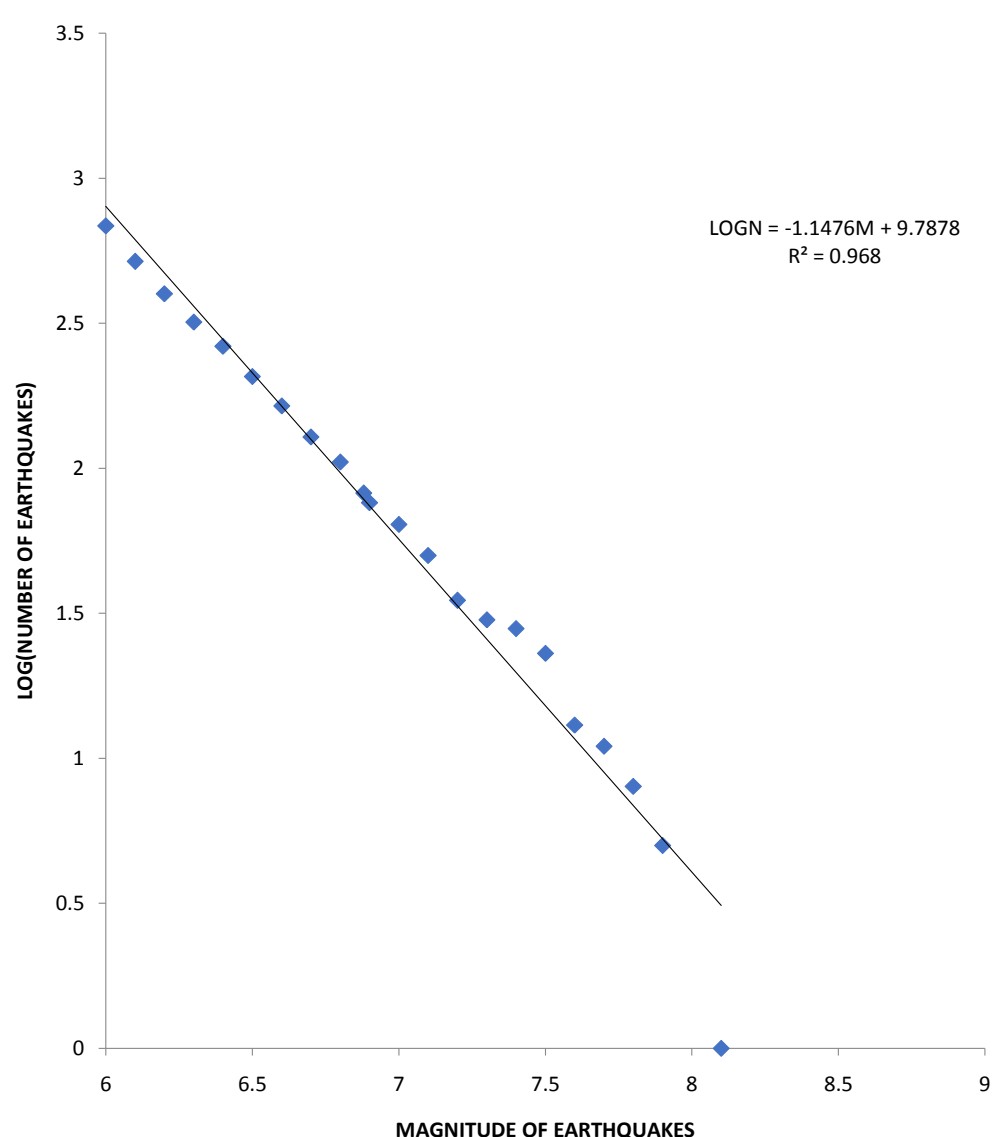


Fig. 7 Frequency Magnitude Distribution of Earthquakes along Southern Hemisphere $5^{o} - 10^{o}$S

LOGN = -0.972M + 8.458
R² = 0.950


Fig. 8  Frequency Magnitude Distribution of Earthquakes along Southern Hemisphere $10^{o} - 15^{o}S$


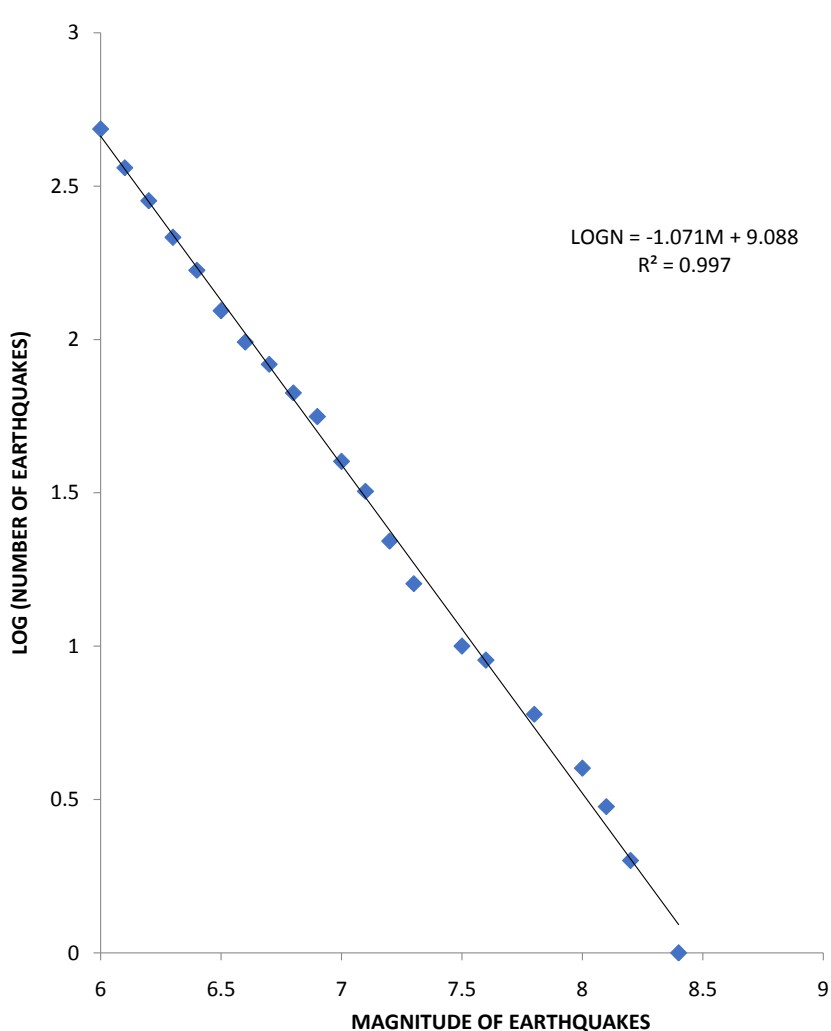


Fig. 9  Frequency Magnitude Distribution of Earthquakes along Southern Hemisphere 15° – 20°S







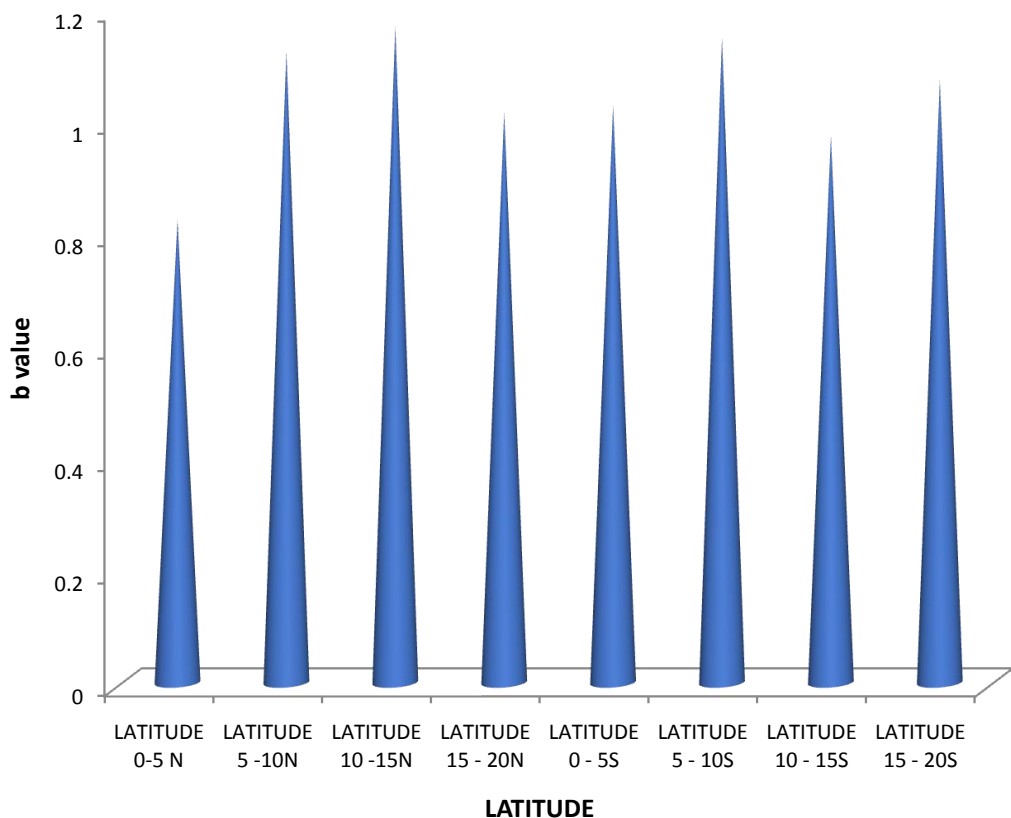


Fig. 10  Distribution of b-value along Northern and Southern Hemisphere



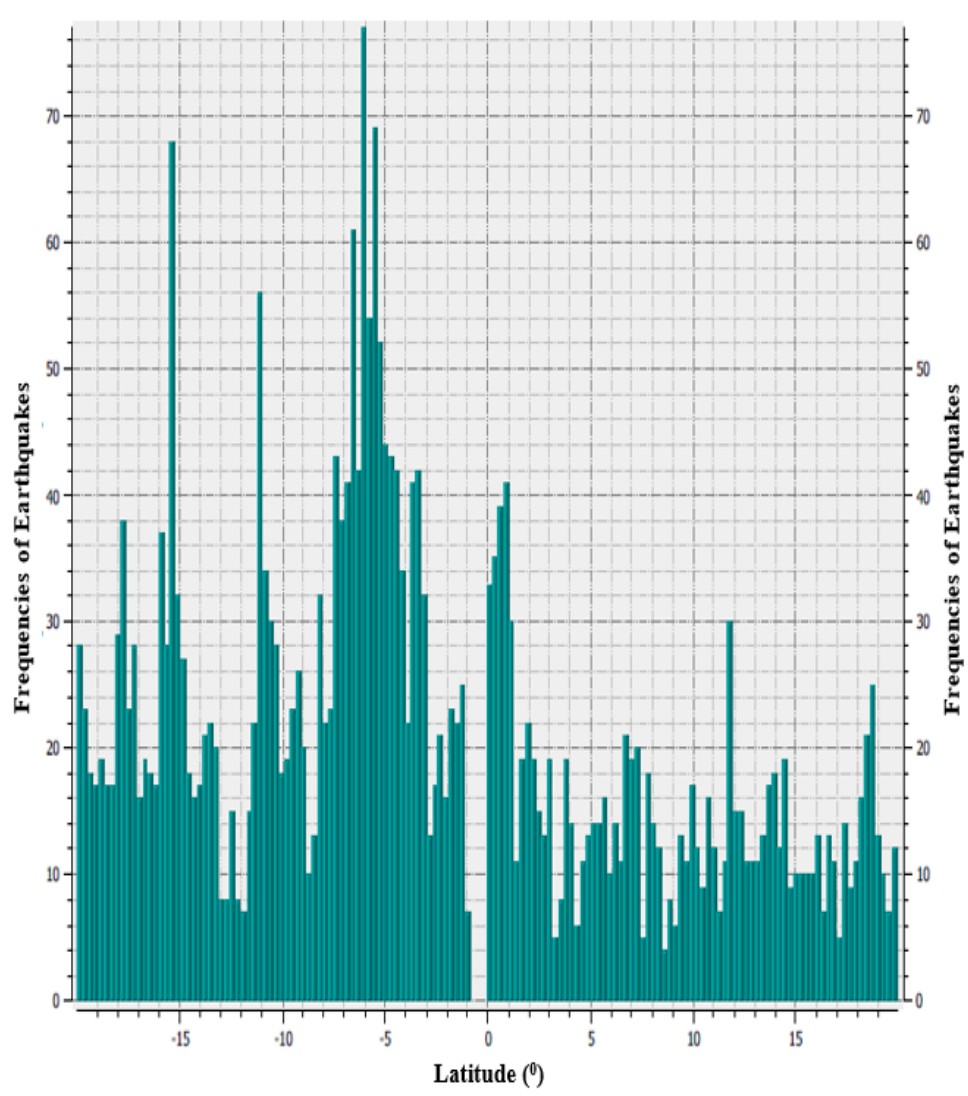


Fig. 11  Distribution of frequency of earthquakes along Northern and Southern Hemisphere



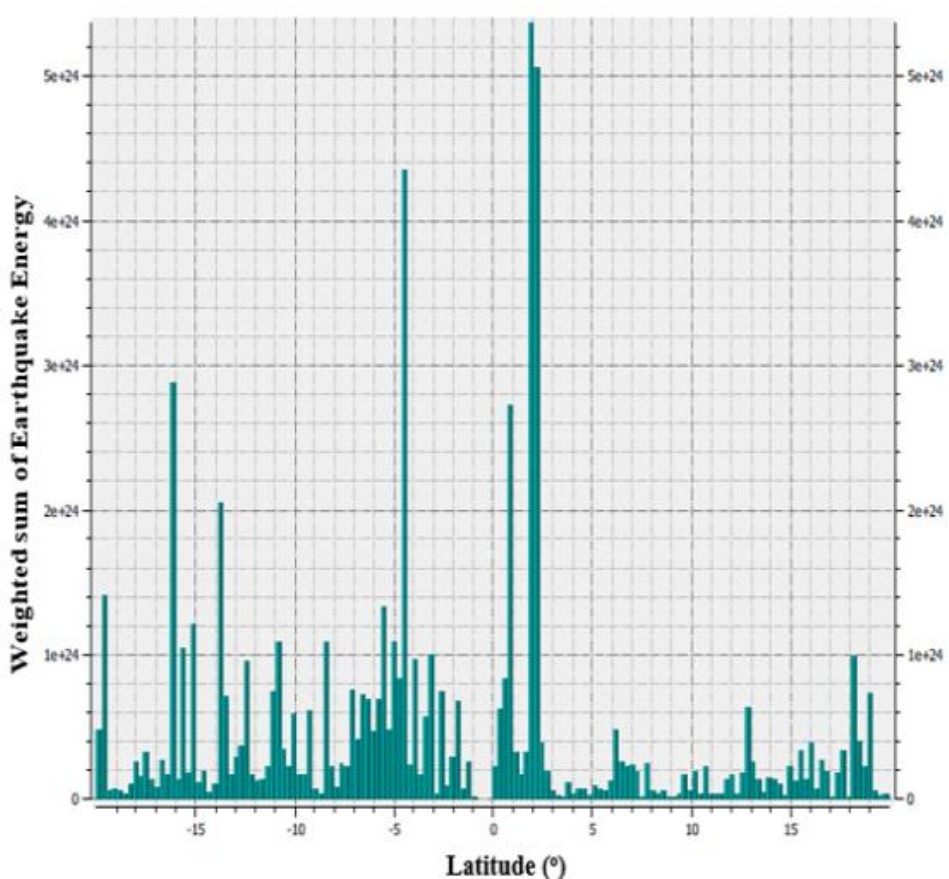


Fig. 12    Distribution of Weighted Sum of Earthquake Energy along Northern and Southern
Hemispheres



Fig. 13a  3D Focal depth distribution of Frequency of Earthquakes along the Latitudinal Zones


Fig. 13b  Focal depth distribution of Earthquake Frequencies along the Northern and Southern
Hemispheres


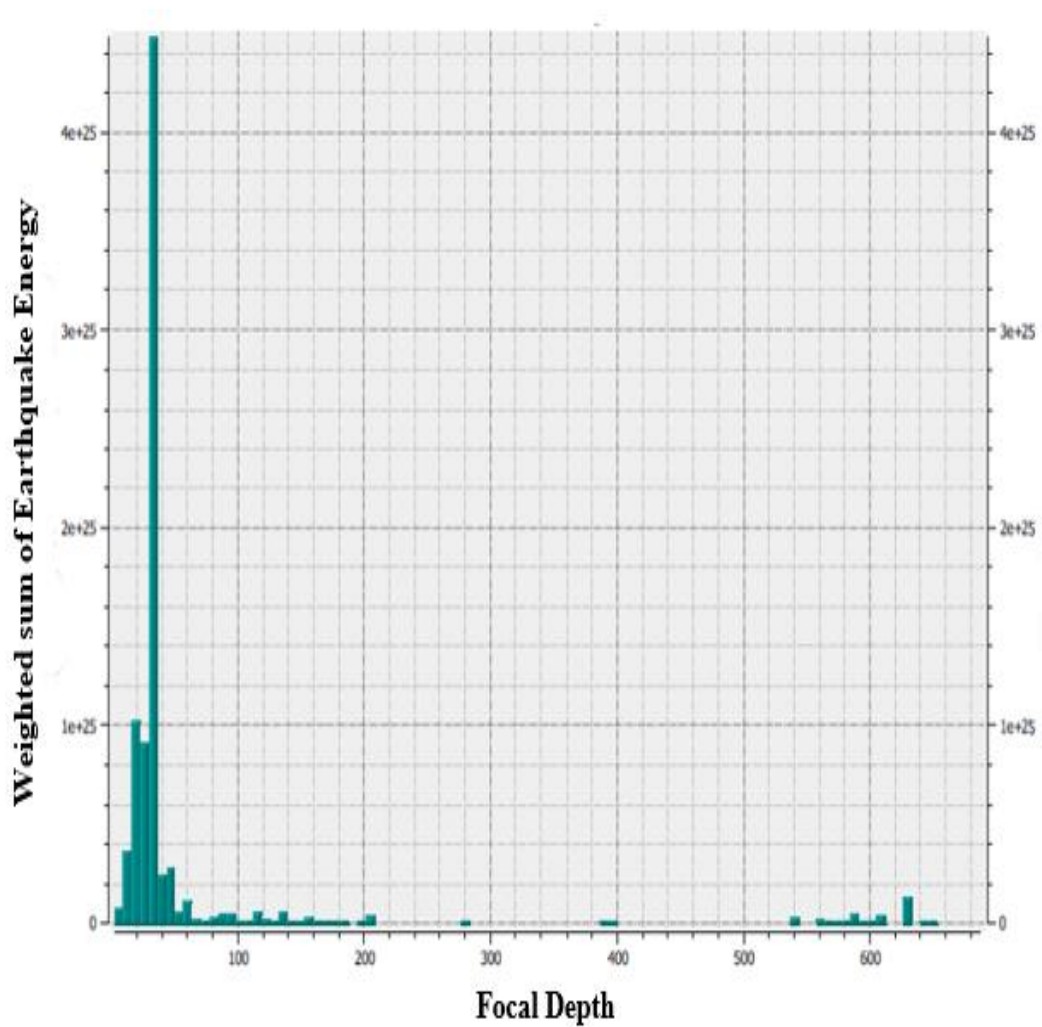


Fig 14   Focal depth distribution of weighted sum of Earthquake Energy along the Northern and
Southern hemispheres





219 Fig 15 Focal depth distribution of magnitudes along the Northern and Southern Hemispheres




Fig. 16  Temporal variation of frequency of Earthquakes along Northern and Southern Hemispheres



## 4. Conclusion

The trend of seismicity along the hemispheres revealed that the rate of earthquake occurrence in the Southern hemisphere is higher than that of the Northern hemisphere. The b-values obtained from this study vary from 0.82 to 1.16. Very low b-values dominated the equatorial region, that is, 5° S to 5° N. This indicates that the level of stress around the equator is relatively high. The maximum earthquake energy of about $4.6 \times 10^{25}$ J was estimated. 70% of the total amounts of energy calculated are housed in the shallow depth, while the intermediate and deep depths share 15% each. The shallow and the deep depths constitute the seismicity regime, as the numbers of events in the intermediate depth are scarce. To further evaluate the number of energies being hosted around the equator, 5° S to 5° N, a maximum of $5.4 \times 10^{24}$ J of earthquake energy was estimated. This indicates that the equatorial regions are characterized by large tectonic stress accumulation. It can be concluded that the lithospheric plates around the equator are unstable. The temporal distribution of seismicity along the hemispheres revealed that the earthquakes increase with time, and decade wise. There is a strong plausibility that the regions around the equator may be prone to disastrous earthquakes in the future.

## Author contribution

OSH, TAA and TEK conceived the study. OSH, TAA and MOA supervised the work. OSH, TAA, MOA, JOA and TSF processed, and anlyzed the data. TEK wrote the first draft. OSH and TAA revised, edited, and searched for the literature to produce the second and final draft. OSH, TAA, MOA, JOA, TSF and TEK gave technical supprot to improve the quality of the paper, read and approved the final draft for submission.

## Competing Interests

We declare that there is no conflict of interest as regards this article.

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
