# Peer review of "Olaide Sakiru Hammed1, Theophilus Aanuoluwa Adagunodo2,\*, Musa Oluwafemi Awoyemi3, Joel Olayide Amosun4, Tokunbo Sanmi Fagbemigun4 and Tobi Ebenezer Komolafe1"

_Natural Hazards and Earth System Sciences, 2020_

## Referee Comment (RC1) · Anonymous Referee #1 · 5 Jun 2020

general comment: The paper by Hammed et al., attempts to estimate the Gutenberg-Richter law b-value, depth distribution and seismic energy release of earthquakes occurred across latitudinal zones, parallel to the equator. In general, the manuscript is way too brief and its content is inadequate to sufficiently cover such a topic. Most important, there are major methodological and conceptual issues leading to ambiguous results and erroneous interpretations. As a result, the manuscript's scientific quality, scientific significance and presentation quality do not meet the expectations of NHESS. Therefore it cannot be accepted for publication.

specific comments: The manuscript is rather short and the language is often plain, needing revision in several points. The quality of figures is also rather poor. The reference list could be updated with more relevant and more recent studies. Even the

title 'Evaluation of global seismicity along Northern 1 and Southern hemispheres' is misleading, since the analysis is focused in a narrow zone (20oS to 20oN) and not to the entire hemispheres. However, linguistic and technical issues are only secondary in comparison with the major methodological and conceptual issues summarized below: 1) Concept. The authors estimate seismicity parameters (b-values, energy release and depth distribution) in 'the study area, which is a strip of width 40o around the globe with the equator at the middle, was subdivided into four regions, each of 5o widths along the Northern and Southern hemispheres' (lines 57-59 of the manuscript). This approach leads to a cascade of conceptual issues which make the study problematic. What is the physical meaning of selecting such areas? A global latitudinal division leads to datasets with events belonging to completely different seismogenic zones, thus demonstrating different properties such as activity rate, magnitude distribution, focal mechanisms, relative plate velocities, stress-strain accumulation, and depth distribution. This horizontal division followed by the authors dissects specific seismotectonic zones with strongly variable properties. For the same reason, technical issues such as hypocentral uncertainties and completeness magnitude are also strongly heterogeneous within each zone. The division and study of smaller and tectonically defined areas (such as the Flinn–Engdahl regions), would be a preferable approach. 2) Methodology: There is no information on how the b-values are calculated. It seems to me that the authors use the least square technique, which has been continuously proven to be inferior to the widely accepted maximum likelihood estimator (e.g. Aki, 1965). The completeness of the datasets is not thoroughly justified as well. How Mc=6.0 arises? Is this level the same for all zones? Is this level homogeneous across each zone? Is this level the same throughout the 55 years of the study period (shown in figure 16)? The authors themselves indicate that 'This deviation could have been due to improved monitoring of the equatorial region' (lines 157-158). Nevertheless, they conclude that 'The temporal distribution of seismicity along the hemispheres revealed that the earthquakes increase with time, and decade wise'. This conclusion is not supported at all by the analysis they present. It is also unclear, which magnitude scale the authors use. Equations 2 and 3 refer to body wave (mb) and surface wave (Ms) magnitudes, however, there is no clear statement which one (or any other) was used for their analysis. Both mb and Ms saturate (∼6.5 and ∼8.0, respectively), therefore they are inappropriate for studying global seismicity with magnitudes up to 9.0.

3) Results: The b-values were found to vary between 0.82-1.16. Even assuming that these values are uniform within each zone and are correctly estimated (see point 2), do they really differ? The authors should provide an analysis to show that the differences of b-values in each zone are statistical significant. The same stands for the seismic activity rates and generally, with any quantitative comparison. According to the authors the smallest b-value and largest energy release occurs in Northern Hemisphere 0o-5oN (figure 2). This is tightly connected with the fact that in this area the two strongest events (M=9.0 and M=8.6) occurred in this area. All other events in all zones have M≤8.5. These M>8.5 are very strong events dominating the energy plot. The authors conclude that 'There is a strong plausibility that the regions around the equator may be prone to disastrous earthquakes in the future' (lines 239-240). This conclusion is definitely not supported by the evidence they provide. Earthquakes with M>8.5 are very rare and the instrumental record does not provide adequate evidence to study how these events occur during centuries, even millennia. The fact that an M>9.0 event has not been recorded during a ∼50 year period does not provide any evidence that such events are not plausible.

technical corrections: Technical corrections are of secondary importance and were not listed in this review.

---

## Referee Comment (RC2) · Anonymous Referee #2 · 16 Jun 2020

General comment: In the manuscript entitled "Evaluation of global seismicity along Northern and Southern hemispheres" by Olaide Sakiru Hammed et al. an attempt is made for the study of the global seismicity by using the earthquake catalog of the Advanced National Seismic System during the period 1963 - 2018. The authors evaluate the Gutenberg Richter b-value parameter and focal depth distribution of earthquake parameters to identify the prominent earthquake-prone zones in the Northern and Southern hemispheres. Although the topic is interesting it is also broad (a comparative study of results additionally obtained from the Centennial earthquake catalog and the GCMT earthquake catalog would make the study more complete). The presentation of the methods is not up to date (see, e.g., Nava, F.A., Márquez-Ramírez, V.H., Zúñiga, F.R. et al. Gutenberg-Richter b-value maximum likelihood estimation and sam-

ple size. J Seismol 21, 127–135 (2017). https://doi.org/10.1007/s10950-016-9589-1 and references therein) while the presentation of the results is poor: For example, the type of the bar chart in Fig.10 is uncommon, what is the definition of the Weighted sum of Earthquake Energy in Figs. 12, 14? What is the meaning of frequency in Fig.11? There are also letters missing in Fig.15 while in Fig.16 I think Frequency is Annual Frequency or Number of earthquakes. Finally, a map of the earthquake epicenters would be much more important for the paper (and would facilitate the relative discussion) than the political map shown in Fig.1. The authors should also check whether they refer to erg or J energy units in various points in their manuscript.

In view of the above, I cannot recommend publication of the present manuscript.

Specific comments: The points raised in my general comments above also include specific points that the authors should consider.

---

## Referee Comment (RC3) · Y. Kamer (Referee) · 19 Jun 2020

The presented manuscript investigates the supposed b-value variation with geographical latitude speculating that low b-values are an indication of high tectonic stress and concluding that there are more earthquakes in the southern than in the northern hemisphere. The paper fails most criteria for a scientific publication, with the exception of a sufficiently clear presentation.

The main problem is that the authors fail to pose a scientific question, a falsifiable hypothesis. Obtaining a data set and applying numerical methods to it will almost always yield results: divide a data set into two, calculate the averages (indeed b-values $\approx$ 1/average magnitude) and you will obtain two different values. Divide the data set

into N sets, you will obtain N different values. Discussing the obtained differences can be regarded as a scientific act only in the presence of a motivated and falsifiable hypothesis that is posed beforehand.

In this regard the presented manuscript is an example of the ongoing crisis is the field of seismology. Kagan highlighted this crisis in his 1999 paper entitled "Is Earthquake Seismology a Hard, Quantitative Science?", arguing that "Higher standards for research in earthquake seismology must be enforced. Authors should adopt a more rigorous style of scientific investigation, and reviewers and editors of geophysical journals should reject manuscripts which do not satisfy the above requirements"

Therefore I recommend the authors to formulate a clear hypothesis, for instance:

-"B-values increase as a function of latitude"

-"Latitudinal b-values variation is significant and therefore provides improved earthquake forecasting performance"

Based on the propose hypothesis the authors should then propose a method to test their hypothesis, for instance:

- Divide the dataset into training and validation and perform a pseudo-prospective test

- Use statistical significance tests to compare the obtained distributions

- Use information criteria to penalize for model complexity (i.e number of subsets) and evaluate the model fit.

I would also like to acknowledge that it is rather unfair that researchers from prestigious institutions often get away with publishing similarly sub-standard research (no hypothesis, no tests), setting a bad example and creating the illusion that such work is acceptable.

With regards to the technical details of the paper, I agree with the points raised by the other reviewers:

- Use Aki's maximum likelihood method to fit b-value

- Bin magnitudes to account for magnitude errors and correct the b-value estimate to account for the binning

- Perform goodness of fit tests that answer the question "how plausible is the proposed b-value as a generating process for the observed magnitudes"

- Perform bootstrapping to evaluate parameter uncertainties

Yavor Kamer

2020.06.19

---

## Referee Comment (RC4) · Matteo Taroni (Referee) · 24 Jun 2020

In this manuscript the authors try to investigate the correlation between the b-value of the Gutenberg-Richter law and the Latitude of the earthquakes. In my opinion they made 3 very important mistakes: 1) b-value must be estimated using the maximum likelihood approach; using the least square approach can lead to a strong bias in the estimation. 2) the magnitude of completeness seems to be underestimated in the first years (see figure 16: very low number of events in the first years). 3) you must test the statistical significance of the differences in the estimated b-values; simply showing the differences without any statistical test do not prove your hypothesis.

Taking into acount this 3 very important mistakes, this manuscript cannot be accepted

for publication.

Other small corrections are of secondary importance and were not listed in this review.

Suggestions to improve the manuscript for a possible re-submission: 1) consider the Global CMT catalog; 2) use the MLE for the b-value (see papers by Ian Kagan and Stefan Wiemer); 3) test the results (see papers by Tokuji Utsu); 4) ZMAP software can help you research (http://www.seismo.ethz.ch/en/research-and-teaching/products-software/software/ZMAP/); 5) take a look to the Community Online Resource for Statistical Seismicity Analysis (http://www.corssa.org/en/home/);

Matteo Taroni

---

## Referee Comment (RC5) · Anonymous Referee #5 · 26 Jun 2020

The authors propose to evaluate and analyze the Gutenberg- Richter b-value parameter and the focal depth distribution with the aim of classifying which zones, between Northern and Southern hemispheres, will be more earthquake-prone. They conclude that the Southern hemisphere will result in a higher rate of earthquake occurrence, and that the lithospheric plates around the equator are unstable.

Despite the argument discussed is of wide interest, this work presents several basic problems that cast doubts upon the results obtained.

The b-value parameter estimation is subject to several sources of bias that would lead to erroneous interpretation in physical terms. For example, the completeness magnitude, the number of events and the binning of magnitudes have been proved to highly influence the estimation of this parameter (see Marzocchi et al., 2019). In the present

paper the authors do not clearly and exhaustively discuss the b-values they obtained, neither how they would be influenced by the potential sources of bias. It would be interesting to compare the estimates presented in the paper with those obtained by the more classical and most used Aki's maximum likelihood method. A detailed discussion about the completeness magnitudes should also be included, since in the present manuscript it is not discussed at all.

The study area considered for the analysis is really very wide, and a segmentation to smaller zones could lead to different results. Furthermore, several geographical regions, scenario of very strong earthquakes, such as Japan and California, are not included. It would be interesting to repeat the analysis also for these latitudes.

The statistical methodologies applied are limited, and a cutting-edge analysis would benefit the robustness of the results obtained.

In my opinion, authors come to poor conclusions that, on their own, do not have a relevance such to justify publication. For all these reasons, I recommend rejection.

References: W. Marzocchi, I. Spassiani, A. Stallone, and M. Taroni. How to be fooled searching for significant variations of the b-value. Geophysical Journal International, 11 2019. ISSN 0956-540X. doi: 10.1093/gji/ggz541. URL https://doi.org/10.1093/gji/ggz541.

---

## Referee Comment (RC6) · Anonymous Referee #1 · 26 Jun 2020

In relation to Referee #5 comments, the authors may also refer to the similar but earlier published work: Leptokaropoulos, K., A. Adamaki, R. Roberts, C. Gkarlaouni, and P. Paradisopoulou (2018), Impact of magnitude uncertainties on seismic catalog properties, Geophys J. Int., 213, 940-951, https://doi.org/10.1093/gji/ggy023.

---

## Referee Comment (RC7) · Yavor Kamer (Referee) · 26 Jun 2020

Dear Anonymous Reviewer,

The paper that you recommend uses Bender's 1983 formula for grouped data. In 2003 Marzocchi and Sandri [1] showed that Tinti and Mulargia's 1987 formula (equation 3.9 and 3.10) is more accurate.

It is unfortunate that a paper investigating systematic biases uses a biased estimator, while being recommended here as a reference to guide others.

Kind regards, Y.Kamer

[1] Marzocchi, W. and Sandri, L., 2009. A review and new insights on the estimation of

the b-value and its uncertainty. Annals of geophysics, 46(6).

[Figure]

The validity of eqs. (3.6) and (3.7) deserves further explanations. In particular, these equations assume that $E(\hat{b}^*) = b^*$ and $E(\hat{b}) = b$, respectively. If we take the expected value of Taylor's expansion around the true value $\mu$ of eqs. (2.3) and (3.1), we see that these assumptions hold only for small deviations of $\hat{\mu}$, *i.e.* for large datasets. Numerical investigations have shown that the biases are negligible for datasets with 50 or more earthquakes.

By comparing eqs. (3.6) and (3.7) we obtain

$$\sigma_{\hat{b}^*} = \left(1 + \frac{\theta_2}{b}\right)^2 \sigma_{\hat{b}} \qquad (3.8)$$

therefore, $\sigma_{\hat{b}^*} > \sigma_{\hat{b}}$. From eqs. (3.5) and (3.8), we can conclude that the true dispersion of the RV $\hat{b}^*(\sigma_{\hat{b}^*})$ increases more than the increase in the estimation of the uncertainty $\hat{\sigma}_{\hat{b}^*}$. In other words, eq. (2.4) provides an underestimation of the true dispersion.

**3.2. *Binned formulas**

After the correction suggested by Utsu (1966), Bender (1983), Tinti and Mulargia (1987) provided formulas to estimate the *b*-value, by properly taking into account the grouping of the magnitudes. Remarkably, besides very few exceptions (*e.g.*, Frohlich and Davis, 1983), these formulas were almost ignored in subsequent applications. We argue that the reasons are mainly of a technical nature. Bender's (1983) formula, for example, can be solved only numerically. Moreover, in her analysis she gave more emphasis to the bias $\theta_1$ introduced by the use of the continuous approximation (eq. (2.3)), concluding that the latter provides almost unbiased estimations of the *b*-value if the magnitude interval for the grouping is $\Delta M = 0.1$.

A definite improvement to the estimation of the *b*-value was provided by Tinti and Mulargia (1987). Their formula reads

$$\hat{b}_{TM} = \frac{1}{\ln(10)\,\Delta M} \ln(p) \qquad (3.9)$$

where

$$p = \left(1 + \frac{\Delta M}{\hat{\mu} - M_{\text{thresh}}}\right) \qquad (3.10)$$

and the associated asymptotic error is

$$\hat{\sigma}_{\hat{b}_{TM}} = \frac{1 - p}{\ln(10)\,\Delta M \sqrt{Np}} \qquad (3.11)$$

where $N$ is the number of earthquakes. In this case, we think the very scarce use of these formulas was probably due to some kind of crypticism of the paper.

**3.3. *Numerical check**

In order to check the reliability of the formulas described above, we simulate 1000 seismic catalogs, for different catalog sizes. The magnitudes $M$ are obtained by binning, with $\Delta M = 0.1$ (as for the instrumental magnitudes), a continuous RV distributed with a pdf given by eq. (2.2); in other words, $M_i$ is the magnitude attached to all the synthetic seismic events with real continuous magnitude in the range $M_i - 0.05 \leq M < M_i + 0.05$.

In fig. 1a,b we report the medians of $\hat{b}^*$, $\hat{b}$ and $\hat{b}_{TM}$ calculated in 1000 synthetic catalogs as a function of the number of data, for the case $b = 1$ and $b = 2$. To each median is attached the 95% confidence interval, given by the interval between the 2.5 and 97.5 percentile. From fig. 1a,b, we can see that the estimation $\hat{b}_{TM}$ (Tinti and Mulargia, 1987) is bias free, also for a small dataset. As regards the continuous formulas, with and without correction (respectively eqs. (3.1) and (2.3)), we can see that the bias $\theta_2$ reported in fig. 1a,b is comparable to the theoretical expectation given by eq. (3.3). The corrected estimation $\hat{b}$ is undoubtedly much closer to the real *b*-value. The slight underestimation of $\hat{b}$ (much less than 1% of the real *b*-value) is due to the bias $\theta_1$ previously discussed (Bender, 1983). Therefore, at least for $\Delta M = 0.1$, $\theta_1$ can be neglected (*e.g.*, Bender, 1983), but $\theta_2$ is certainly relevant.

In order to evaluate the reliability of the estimations of the uncertainty, it is necessary to compare each estimation with the true dis-

**Fig. 1.**

[Figure]

---

## Referee Comment (RC8) · Yavor Kamer (Referee) · 26 Jun 2020

Dear Angeliki K. Adamaki,

As you can see in the attached figure (taken from MS2009) the correction factor that you have used (3.1) gives biased estimates of the b-value (dotted lines). The correction factor proposed by Tinti & Mulargia (3.9, 3.10) should be used (solid line).

You can also read it from the text "From fig. 1a,b, we can see that the estimation bTM (Tinti and Mulargia, 1987)is bias free, also for a small dataset." The conclusion text you referred to is about the general use of correction for grouped data.

Kind regards, Y. Kamer

[Figure]

[Figure]

ed lines), $\hat{b}$ (dotted lines) and $\hat{b}_{\text{TM}}$ (thicker solid line)

**Fig. 1.**

[Figure]

---

## Short Comment (SC1) · 26 Jun 2020

Dear Dr Kamer, if I understand all previous comments correctly, the article by Leptokaropoulos et. al, 2018 calculates the b-value using a correcting factor for the binning (eq. 4 in their article). The same formula is tested in the article you mention by Marzocchi and Sandri 2009 (eq. 3.1), and the authors conclude that "the continuous formula with a small correction to take into account the binned magnitudes drastically reduces the biases of the b-value and of its uncertainty" referring to the same equation Leptokaropoulos et. al, 2018 have used.

Leptokaropoulos, K., A. Adamaki, R. Roberts, C. Gkarlaouni, and P. Paradisopoulou (2018), Impact of magnitude uncertainties on seismic catalog properties, Geophys J.

Int., 213, 940-951, https://doi.org/10.1093/gji/ggy023

Marzocchi, W. and Sandri, L., 2009. A review and new insights on the estimation of the b-value and its uncertainty. Annals of geophysics, 46(6), https://doi.org/10.4401/ag-3472

---

## Referee Comment (RC9) · Anonymous Referee #1 · 2 Jul 2020

Dear Dr Kamer, Thank you very much for your comment, however what is really unfortunate is that we need to argue for or against already published (and peer-reviewed) articles, while our goal here is to provide feedback to the authors that submitted an article for review. Nevertheless, a fruitful discussion and exchange of scientific opinions on observations and published results is always welcome in order to investigate deeper (and objectively) what new (or old in that particular case) approaches can offer or probably have missed. Leptokaropoulos et al. (2018) use synthetic datasets in order to investigate biases, meaning that even a biased estimator (as you suggest) is subjected to investigation as well. Nonetheless, as shown in table 2, the selected estimator is not biased at all (giving values best=0.99 and best=1.00 DM=0.1, while b=1.00,), meaning that within the constraints of the given study it is pretty accurate.

[Figure]

The introduction of DM/2 term in Aki (1965) maximum likelihood estimator is a widely used b-value estimation technique and its efficiency has been also verified by many published papers (e.g. Marzocchi et al. 2019 and references therein), stating that magnitude binning introduces significant b-value biases for DM>0.25, while being negligible for DM=0.1. In order to guide the authors, I would suggest them to dig deeper into bibliography and consider all reviews that have been provided here, in order to improve their study in the best possible way given the current knowledge background.

---

## Referee Comment (RC10) · Yavor Kamer (Referee) · 2 Jul 2020

Dear Anonymous,

I agree with you that the effect of using an inappropriate binning correction diminishes as the bin size is reduced. Magnitudes are binned assuming that the magnitude errors are smaller than the bin size. In other words, we are trying to account for magnitude errors by binning.

In a global application, which is the topic here, magnitude errors will sometimes exceed even M0.5. Thus I hope it is clear for the authors why a proper magnitude binning and its correction is essential. The results that you referred to (Table 2 of Leptokaropoulos 2018) are derived for "noise-free" catalogs, hence I don't think they are relevant for the

discussion here.

On a different note, I fail to see why you have trouble conceding that the Tinti & Mulargia formula is demonstrably more accurate. I also didn't know about it, a peer of mine reviewed me and I learned. That's how science works. If you want to discuss further, feel free to contact me in person yaver.kamer@gmail.com

Kind regards, Y. Kamer

---

## Author Comment (AC1) · 21 Jul 2020

We appreciate you for your contribution.

All your comments are noted, and would be integrated in our research for re-assessment.

Thank you.

---

## Author Comment (AC2) · 21 Jul 2020

We appreciate you for your contributions.

All your comments are noted, and would be integrated in our research for re-assessment.

Thank you.

---

## Author Comment (AC3) · 21 Jul 2020

We appreciate you for your contributions.

All your comments are noted, and would be integrated in our research for re-assessment.

Thank you.
* * *

---

## Author Comment (AC4) · 21 Jul 2020

We appreciate you for your contributions.

All your comments are noted, and would be integrated in our research for re-assessment.

Thank you.

---

## Author Comment (AC5) · 21 Jul 2020

We appreciate you for your contributions.

All your comments are noted, and would be integrated in our research for re-assessment.

Thank you.

---

## Author Comment (AC6) · 21 Jul 2020

Thank you once again Sir.
* * *

---

## Author Comment (AC7) · 21 Jul 2020

We appreciate you for your contributions.

All your comments are noted, and would be integrated in our research for re-assessment.

Thank you.

---

## Author Comment (AC8) · 21 Jul 2020

We appreciate you for your contributions.

All your comments are noted, and would be integrated in our research for re-assessment.

Thank you.

---

## Author Comment (AC9) · 21 Jul 2020

We are grateful for this information.
* * *